# The Thermomechanical Finite Element Analysis of 3003-H14 Plates Joined by the GMAW Process

**Maribel Hernández [1], Ricardo R. Ambriz [1,*], Christian García [2] and David Jaramillo [1]**

[1] Instituto Politécnico Nacional CIITEC-IPN, Cerrada de Cecati S/N Col. Sta. Catarina, Azcapotzalco 02250, Ciudad de México C.P., Mexico; h.guerrero.maribel@gmail.com (M.H.); djvigu@gmail.com (D.J.)

[2] Facultad de Ingeniería, Universidad Autónoma de San Luis Potosí, San Luis Potosí 78290, Mexico; dosnoch@gmail.com

* Correspondence: rrambriz@ipn.mx; Tel.: +52-555-729-6000

**Abstract:** The gas metal arc welding (GMAW) process was used to weld 3003-H14 plates under restricted and unrestricted thermal expansion. Experimental and numerical analysis were conducted to determine the relation between weld thermal cycles and residual stresses. A customized data acquisition system with K-type thermocouples was used to measure the weld thermal cycles, while residual stresses were determined by the hole drilling method. Thermo-mechanical simulation models for the two restricted conditions were implemented from the experimental data obtained. A double ellipse heat distribution geometry was used to model the heat moving source by using the finite element method. Thermal rates and peak temperatures were approximated by the finite element model with 2% difference, with respect to the experimental weld thermal cycles. Longitudinal and transverse normal residual stresses determined by the finite element model showed a good comparison with experimental measurements. The larger residual stresses were in the transverse direction for both clamping conditions, which indicated that working loading paths along the lateral direction of the welded plate are more influenced by the post-welding residual stresses.

**Keywords:** aluminum 3003-H14; GMAW; heat moving source; residual stresses; finite element analysis

---

## 1. Introduction

3XXX aluminum alloys are commercial wrought materials, which have Mn as a primary element alloy. These materials are non-heat treatable by precipitation. The principal hardening mechanism is provided by cold working as specified by the -H14 designation [1]. This cold working condition indicates 50% of strain hardening, which allows to obtain a minimum yield strength at 0.2% strain of 115 MPa and an elongation of 8% [2]. This alloy presents an excellent workability, weldability, and corrosion resistance, and is industrially used to manufacture products where moderate strength is required.

Several welding processes have been used to weld aluminum alloys. The gas metal arc welding (GMAW) process is one of the most common processes used for industrial applications. This process uses a constant voltage power source, a filler wire material, and a shielding gas (inert gases for aluminum alloys). The GMAW process generates an intense power source, which is used to perform the weld. The first studies related to a heat moving source were performed by Rosenthal [3,4]. These works laid the mathematical and scientific basis for understanding a heat moving source and its application to welding and cutting processes. For instance, Rosenthal in [4] established that by means of a single formula it is possible to predict the time and rate of cooling with a fairly good accuracy for a wide variety of thicknesses of steel, ranges of temperature, and welding conditions. Two mathematical models (thin and thick plate models) were derived to study the heat conduction transfer phenomenon.

For decades, those models have been used successfully and several other models based on Rosenthal's theory have been developed [5]. These models have been extrapolated to the finite element calculation. Goldak et al. [6] developed a finite element model for welding heat sources to compute the thermal history, which considers a double ellipsoidal geometry for the heat input. This model has been used successfully, and it has been shown to be more accurate than the disc model proposed previously by Pavelic et al. [7].

On the other hand, the high thermal gradients induced by the welding heat source and the thermal history produce welding residual stresses due to the expansion–contraction phenomenon during the heating and cooling process. This induces a heterogeneity of the residual stresses in all directions. The longitudinal residual stresses are related to the longitudinal contraction and expansion of the weld bead. Whereas, transverse residual stresses are generated in accordance with the mechanism of a weld bead which contracts transversely, especially when the plates are restrained. Residual stresses can be experimentally measured by different techniques depending on the method, penetration, spatial resolution, and accuracy [8]. These techniques include destructive and non-destructive procedures, i.e., hole drilling, X-Ray diffraction, neutron diffraction or ultrasound. For instance, the hole drilling is a standardized test method to determine residual stresses near the surface of an isotropic linear-elastic material [9], which involves attaching a strain rosette to the surface, drilling a hole at the geometric center of the rosette and measuring the resulting relieved strain. However, to determine the residual stress distribution by this procedure is necessary to use many strain sensors, which is expansive and impractical in some cases, because the specimen needs to be drilled and possible destroyed. In contrast, if the origin of the residual stresses is clearly identified, the finite element method can be used to determine the stress distribution with an acceptable accuracy. In this context, the residual stresses generated by a heat welding source have been analyzed by using the finite element method [10–15]. Cañas et al. [10] carried out a simplified numerical analysis to determine residual stresses in aluminum welded plates by using a plane stress finite element model. They reported that numerical results are in very good agreement with experimental measurements obtained by the hole drilling method. Recently, Lu et al. [15] performed a three-dimensional finite element model to determine longitudinal and transverse residual stresses in aluminum alloy (A7N01 alloy) welded joints performed by GMAW. They found that longitudinal and transverse residual stresses were generally tensile stresses in the weld areas and the adjacent heat affected zone. Also, it was reported that transverse residual stress was considerably smaller than longitudinal residual stress.

This article proposes a finite element model to evaluate the thermo-mechanical phenomenon of the 3003-H14 aluminum alloy plates welded by GMAW. In addition, an experimental welding procedure was conducted to compare with the finite element results. For the finite element model, the double ellipse geometry for the heat input model was considered to obtain the temperature distribution along the welded plates. This model considers the conduction and convection phenomena. Two different clamping welding conditions were used to determine the residual stress distribution (longitudinal and transverse stresses) along the welded plates. Weld thermal cycles and residual stresses were experimentally measured by thermocouples and strain rosettes for comparison against the numerical results.

## 2. Materials and Methods

### 2.1. Materials, Welding, and Experimental Measurements

A commercial cold worked plate of 3003-H14 aluminum alloy (Al-Mn) with 3.5-m length, 1.22-m width, and 6.6-mm thickness was used. Plates were cut out to the dimensions shown in Figure 1 and a single V-groove joint preparation was manufactured for the welding process. The width of the plates was coincident with the rolling direction and perpendicular to the welding heat source. A semiautomatic gas metal arc welding process (GMAW) and an ER4043 filler material (1.2 mm in

diameter) was used to deposit a weld bead. The chemical composition of the base and filler materials is shown in Table 1.

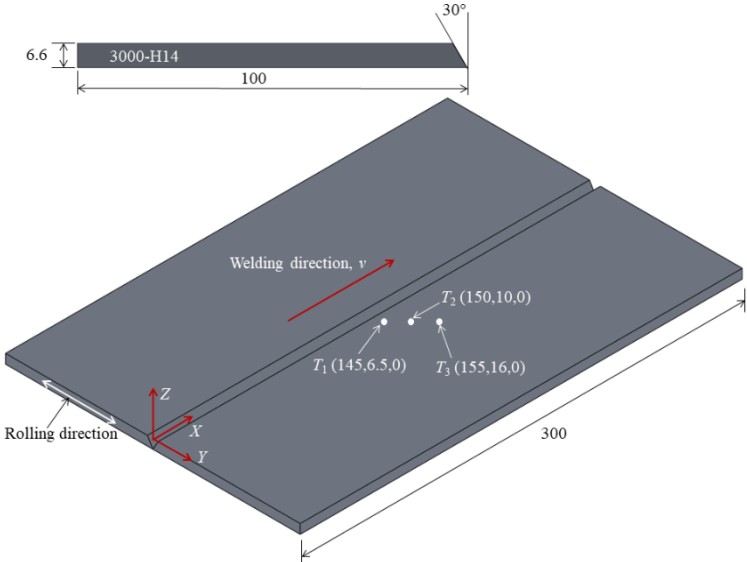

**Figure 1.** Single V-groove join preparation for the 3003-H14 aluminum alloy plates, showing the welding direction and thermocouples ($T_1$, $T_2$, and $T_3$) location according to the global coordinate system ($X$, $Y$, $Z$). Dimensions are in mm.

**Table 1.** Chemical composition of the materials used (weight percent) [2,16].

| Material | Si | Fe | Cu | Mn | Zn | Mg | Ti | Others | Al |
|---|---|---|---|---|---|---|---|---|---|
| 3003-H14 [2] | 0.6 | 0.7 | 0.05–0.20 | 1.0–1.5 | 0.10 | - | - | 0.15 | Balance |
| ER4043 [16] | 4.5–6.0 | 0.8 | 0.30 | 0.05 | 0.10 | 0.05 | 0.20 | 0.15 | Balance |

Argon was used as shielding gas at a flow rate of $16.7\,\text{L min}^{-1}$. The welds were performed by using direct current positive electrode (DCPE) with the following welding parameters: voltage $V = 24$ V, current $I = 156$ A, and travel speed $v = 4\,\text{mm s}^{-1}$. According to the welding parameters, and considering a thermal efficiency of 90% for the GMAW process, the heat input provided during welding was $Q = 842.40\,\text{J mm}^{-1}$.

Microhardness and tensile tests were performed from the original plate. For the microhardness measurements, an indentation load of 0.1 kg (0.981 N) with a dwell-time of 15 s was used. Ten indentations were taken to determine the mean hardness value. Standard dog-bone tensile specimens were machined according to the ASTM B557 standard (subsize specimen).

In situ weld thermal cycles produced by the GMAW process were registered by means of K-type thermocouples, located on the surface of one plate as indicated by the coordinates shown in Figure 1. An approximation of the transient heating period (pseudo steady-state condition) was determined according to Grong [5]. A data acquisition module (NI 9213) coupled directly to each thermocouple was used for the condition and acquisition of the temperature signal. These signals were digitalized by using a LabVIEW program connected to a computer, at a sampling frequency of 75 Hz.

Two different clamping conditions of the plates were used during the GMAW process to generate different residual stress distribution. Low residual stresses were developed in a set of plates welded with no mechanical restriction (Figure 1), meanwhile another set of plates were restricted as shown in Figure 2. Constraint of the aluminum plates was done, by fixing it with bolts to a stainless steel backing plate (Figure 2b). A torque of 80 N m was applied on each bolt.

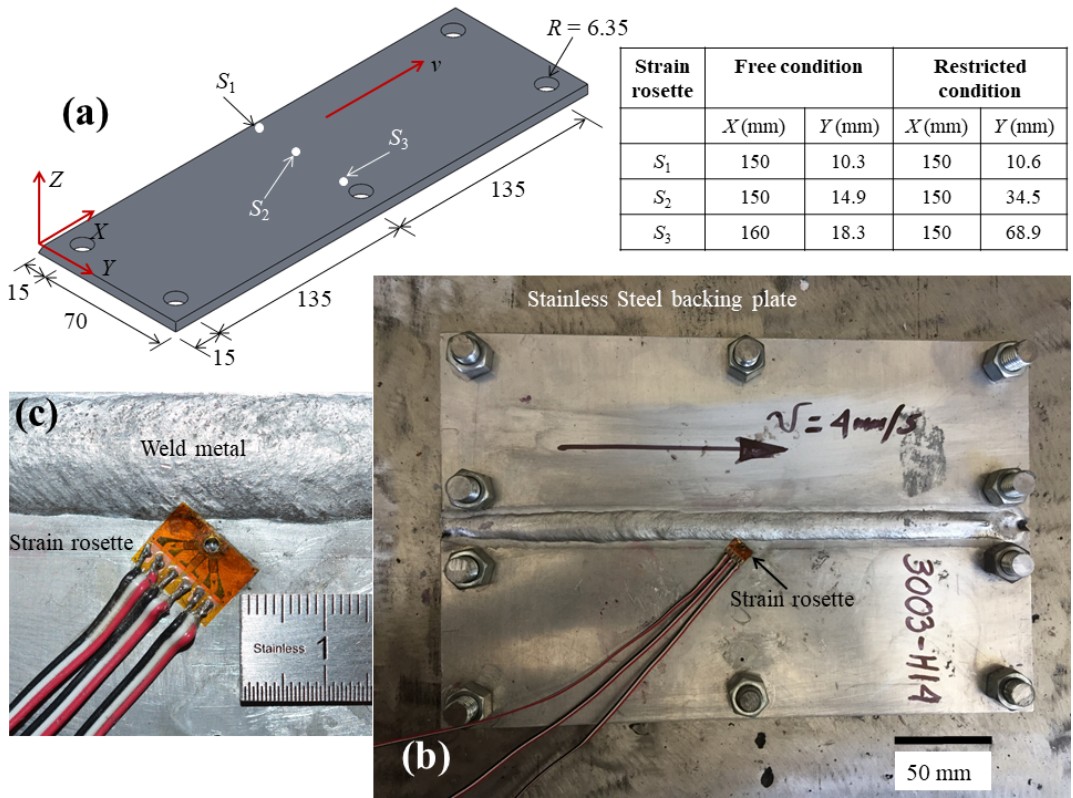

**Figure 2.** (**a**) Schematic representation of a 3003-H14 aluminum plate showing the holes for the displacement restriction as well as the strain rosettes ($S_1$, $S_2$, and $S_3$) location, (**b**) welded plates appearance, and (**c**) close-up of the strain rosette located close to the weld metal. Units in Figure 2a,c are in mm.

The effect of the heat generated by the welding process on microhardness was measured. Five microhardness profiles along a line of 50 mm in length (25 mm from the center of weld metal (WM) to each side) were obtained. The indentations were performed at a separation of 500 µm. The microhardness lines were placed along the thickness of the welding profile with a separation of 1.3 mm from each other.

Hole-drilling method (ASTM E837) [9] was used to measure the residual stresses generated by the welding process. Figure 2a shows the strain sensors location. CEA-13-062MU-120 micro-measurements strain gauges were used. A drill of 1.62 mm in diameter with a RS-200 milling guide was used. The drilling was carried out in 0.1 mm incremental depth steps up to a maximum depth of 1 mm. Throughout the drilling process a VISHAY P3 strain indicator (micro-measurements brand fabricated in Vishay Micro-Measurements, Raleigh, NC, USA) was used to acquire the signals of the rosettes. The Residual stresses were computed by means of Hole-Drilling Residual Stress Calculation Program Version 3.21.

## 2.2. Finite Element Model

Experimental thermal cycles determined by thermocouples as well as the residual stresses measured by the hole drilling method were conducted. The finite element method (FEM) at the ANSYS APDL platform (academic version 19.0) was used to approximate the thermal cycles and residual stress fields generated by the welding process. Table 2 shows the thermophysical properties used for the finite element analysis.

**Table 2.** Thermophysical properties used for the finite element analysis [17–21].

| | | | **Aluminum** | | | |
|---|---|---|---|---|---|---|
| $T$, °C | $C_p$ J kg$^{-1}$ °C$^{-1}$ | $k$ W m$^{-1}$ °C$^{-1}$ | $\rho$ kg m$^{-3}$ | $\alpha$ × 10$^{-6}$ m$^2$ s$^{-1}$ | $\alpha_t$ × 10$^{-6}$ °C$^{-1}$ | $Pr$ |
| 25 | 900 | 141 | 2720 | 57.5 | 23.2 | - |
| 100 | 940 | 148 | 2706 | - | 23.2 | - |
| 200 | 990 | 156 | 2685 | 58.5 | 24.1 | - |
| 300 | 1004 | 158 | 2662 | 59.0 | 25.1 | - |
| 400 | 1066 | 167 | 2638 | 59.5 | - | - |
| 500 | 1111 | 174 | 2611 | 60.0 | - | - |
| 617 | - | 183 | 2583 | 61.0 | - | - |
| 656 | 1220 | 61 | 2572 | 21.0 | - | - |
| 700 | 1220 | 61 | 2388 | 21.0 | - | - |

| | | | **Filler Metal** | | | |
|---|---|---|---|---|---|---|
| $T$, °C | $C_p$ J kg$^{-1}$ °C$^{-1}$ | $k$ W m$^{-1}$ °C$^{-1}$ | $\rho$ kg m$^{-3}$ | $\alpha$ × 10$^{-6}$ m$^2$ s$^{-1}$ | $\alpha_t$ × 10$^{-6}$ °C$^{-1}$ | $Pr$ |
| 25 | 860 | 139 | 2690 | 60.0 | 22.1 | - |
| 100 | 910 | 152 | 2679 | 62.0 | 22.1 | - |
| 200 | 960 | 164 | 2662 | 64.0 | 23.7 | - |
| 300 | 980 | 169 | 2643 | 65.0 | - | - |
| 400 | 1100 | 161 | 2622 | 59.0 | - | - |
| 500 | - | - | 2600 | - | - | - |
| 573 | 1190 | 147 | 2482 | 50.0 | - | - |
| 600 | 1190 | 64.5 | 2473 | 21.7 | - | - |
| 700 | 1190 | 66.5 | 2417 | 23.0 | - | - |

| | | | **Ar** | | | |
|---|---|---|---|---|---|---|
| $T$, °C | $C_p$ J kg$^{-1}$ °C$^{-1}$ | $k$ W m$^{-1}$ °C$^{-1}$ | $\rho$ kg m$^{-3}$ | $\alpha$ ×10$^{-6}$ m$^2$ s$^{-1}$ | $\nu$ × 10$^{-6}$ m$^2$ s$^{-1}$ | $Pr$ |
| 25 | 521.2 | 0.0178 | 1.622 | 21.5 | - | 0.69 |
| 100 | 520.9 | 0.0225 | 1.218 | 35.46 | - | 0.69 |
| 500 | 520.0 | 0.0265 | 0.973 | 52.37 | 33.7 | 0.66 |
| 800 | 520.0 | 0.0369 | 0.608 | 116.7 | 46.6 | 0.66 |
| 1000 | 520.0 | 0.0427 | 0.487 | 168.6 | 42.7 | 0.66 |
| 1500 | 520.0 | 0.0551 | 0.324 | 327.0 | 54.2 | 0.67 |

By assuming that heat input induced by the welding process tends to produce a homogeneous thermal distribution, it was decided to use a half model with symmetric conditions. The dimensions of the aluminum welded plates were considered to create a three-dimensional finite element model. To represent the WM, heat affected zone (HAZ) and base metal (BM), three different volumes were considered. A hexahedral eight nodes finite element (SOLID70) with one degree of freedom (temperature) was selected for the thermal transient analysis. For the structural analysis, a hexahedral finite element (SOLID185) with three degree of freedom by node (displacements in the nodal *X*, *Y*, and *Z* directions), were used.

Figure 3 shows the three-dimension finite element mesh. Since, the dimensions of plates are similar, only the finite element model for the restricted plates is shown. The mesh grows coarser as a function of the distance from the WM along the width, i.e., the coarser mesh was built by finite elements of 5.0-mm length and 1.0 mm for width and thickness, whereas the finer mesh has elements of 0.5-mm length and 1.0 mm for the width and thickness, with a geometry as shown in Figure 3. A total of 93,600 finite elements and 108,360 nodes were used to represent the finite element model for the plates without restriction, whereas for the plates restricted by the fasteners bolts, 92,200 finite elements and 107,640 nodes were needed.

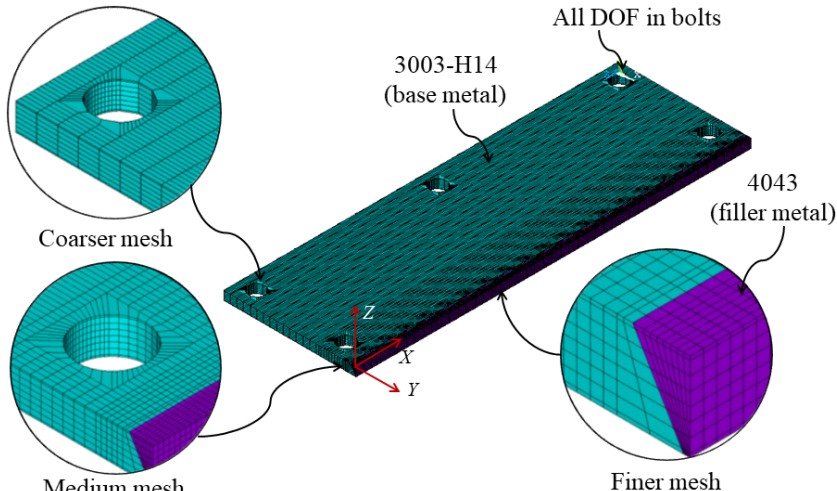

**Figure 3.** Three-dimensional mesh used for the thermo-mechanical analysis.

The following three considerations were used for the thermal transient finite element analysis: (i) The heat input is dissipated by conduction and convection through backing plate, welded plates and the interaction between surfaces with natural airflow at ambient temperature. (ii) The heat flux was quantified from the heat input by using the double ellipse distribution (Figure 4) [6,22]. (iii) The latent heat from liquid-solid transformation was not considered.

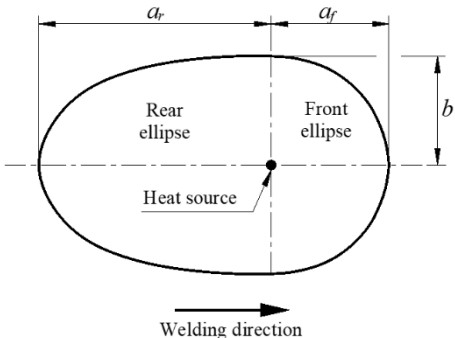

**Figure 4.** Schematic representation for the double ellipse heat source configuration, reproduced from [22] with permission from Elsevier, 2020.

Thus, considering the geometry of the welding pool (Figure 4), the heat distribution of the front and rear ellipse was determined by the following expressions:

$$q(x,\ y,\ t)\frac{6f_fQ}{a_fb\pi}\exp\left[-\frac{3x^2}{b^2}-\frac{3y^2}{a_f^2}\right] \tag{1}$$

$$q(x,\ y,\ t)\frac{6f_rQ}{a_rb\pi}\exp\left[-\frac{3x^2}{b^2}-\frac{3y^2}{a_r^2}\right] \tag{2}$$

In general, the front temperature gradient $(f_f)$ is shorter than the rear half $(f_r)$. Hence, the heat power density in the rear and front ellipses are represented by the fractions $f_f$ and $f_r$ of heat deposited given by:

$$f_f + f_r = 2 \tag{3}$$

From experimental measurements the following values were found: $a_r$ = 9.0 mm, $a_f$ = 8.0 mm, $b$ = 8.22 mm (Figure 7), $f_f$ = 0.94, and $f_r$ = 1.06. Figure 5 shows the surface heat distribution for the

double ellipse model, as used, to determine the heat applied on each specific area on the surface of the finite element model.

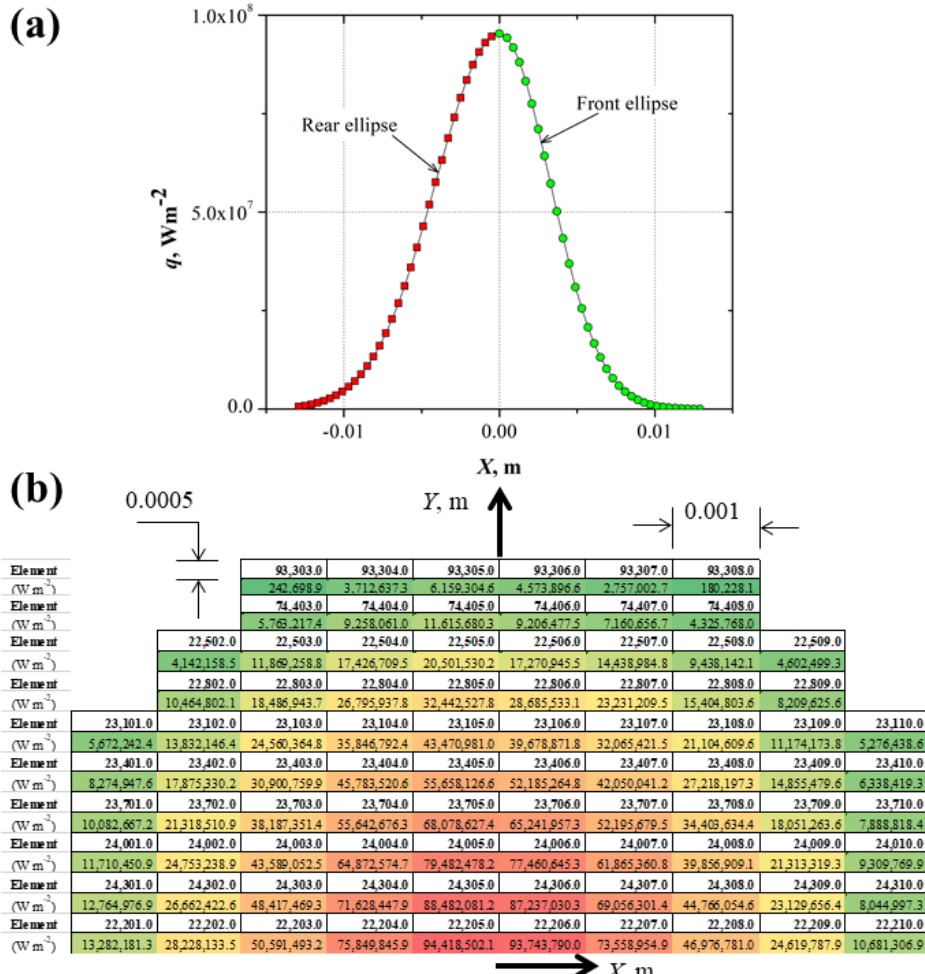

**Figure 5.** Heat flux distribution according to the double ellipse model: (**a**) heat flux as a function of distance and (**b**) heat flux applied to each finite element.

The heat flux distribution (Figure 5a) was applied on the surface of the finite element model. Figure 5b shows the heat flux calculated for each element, corresponding to a loading thermal step of 0.25 s (total time of the heat welding source was 75 s and travel speed of 4 mm s⁻¹). An initial temperature of 25 °C for the aluminum and backing plate was stablished. The convective effect during the displacement of the heat welding source generated by the shielding gas was considered. It is to say, a convective coefficient $h$ was applied at the rear heat source. To determine the $h$ value close to the WM the following equations were used:

$$N_u = 0.453\left(R_e^{0.5} P_r^{1/3}\right) \tag{4}$$

$$R_e = \frac{vL_c}{\nu} \tag{5}$$

$$h = \frac{k}{L_c} N_u \tag{6}$$

where $N_u$ is the Nusselt number, $R_e$ is the Reynolds number, $v$ is the flow speed, $L_c$ is a characteristic linear dimension, $\nu$ is the kinematic viscosity, and $k$ is the thermal conductivity.

The properties of the fluid for the boundary limit were determined by the boundary temperature $T_f$, which is the arithmetic mean value between the surface temperature $T_s$ and the fluid temperature $T_\infty$:

$$T_f = \frac{T_s + T_\infty}{2} \qquad (7)$$

To determine the $h$ value between aluminum and backing plates, the heat sink effect generated by the backing plate was considered:

$$Q = hA_s(T_\infty - T_s) = \frac{kA_s}{L_c}(T_1 - T_2) \qquad (8)$$

The thermal transient finite element model was solved by implementing a thermal loading steps program to apply the heat flux as a function of the heat source displacement. Later, the cooling effect as a function of the cooling time (1200 s) was determined.

The transient thermal results were used to couple with the structural analysis and to determine the residual stresses generated by the thermal effect as well as the restriction conditions. For the structural analysis, the true stress–strain behavior of the materials was considered, i.e., elastic–plastic with a kinematic hardening behavior. The thermal effect generated by the welding process was considered as the only source of displacement (load), whereas the boundary conditions were fixed by the free degree of freedom of the plates (no restriction), and the all displacement restriction for the plates by means of bolts.

## 3. Results and Discussion

### 3.1. Materials and Welding

Figure 6 shows the true stress–strain behavior of the materials used. In the case of the ER4043 alloy (filler metal), the results were taken from a previous work [23]. From Figure 6 it is possible to observe that Young modulus tends to be similar for both materials. However, the plastic region exhibits an important difference, i.e., the yield strength at the 0.2% offset strain for the ER4043 is roughly 20 percent higher than that for the 3003-H14, in contrast, the hardening capacity for the 3003-H14 alloy is almost three times larger than the 4043 alloy. Tensile mechanical properties summary is shown in Table 3. Young modulus, yield strength and tangent modulus were implemented to represent the kinematic hardening mechanical behavior for each material to be used in the finite element model.

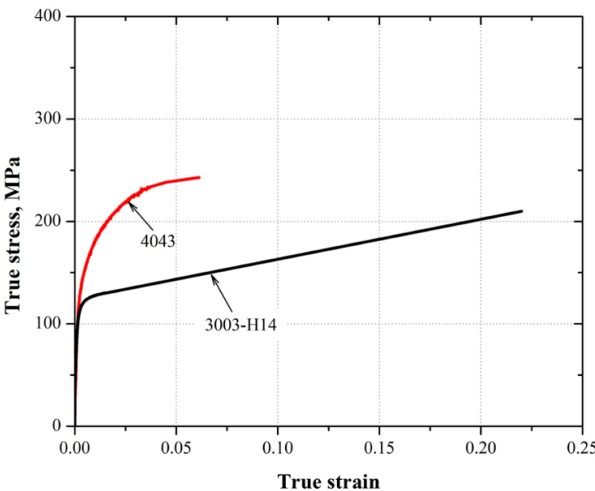

**Figure 6.** True stress–strain behavior for the 3003-H14 and 4043 aluminum alloys, data from [23].

**Table 3.** Tensile mechanical properties of the materials used for the finite element computation.

| Material | $E$ (GPa) | $\sigma_{0.2}$ (MPa) | $E_T$ (MPa) | $H$ (MPa) | $n$ | $\nu$ |
|---|---|---|---|---|---|---|
| 3003-H14 | 64 | 126 | 464 | 510 | 0.586 | 0.33 |
| 4043 [23] | 68 | 151 | 379 | 463 | 0.200 | 0.33 |

Figure 7 shows the welding profile obtained for the 3003-H14 aluminum welded plates. This profile was obtained from the pseudo-steady state region, and prepared metallographically to observe the macrostructural features.

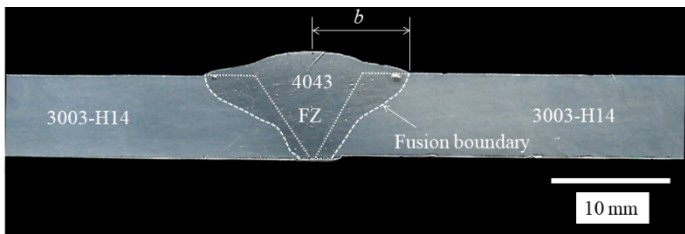

**Figure 7.** Welding profile obtained for the 3003-H14 welded plates.

From Figure 7, it is possible to observe a complete joint penetration as well as an acceptable symmetry of the welding profile geometry. Porosity formation in the WM, attributed to the rapid solidification process as well as to the hydrogen affinity in liquid aluminum was observed. An effective fusion radius ($b$) of approximately 8.22 mm was determined from the welding profile. This value was used to determine the heat flux distribution by using Equations (1) and (2).

The heat effect induced to the aluminum plates by the welding process was evaluated by microhardness measurements (Figure 8). Figure 8 shows the hardness behavior along the welding profile. It is observed that WM (4043 alloy) is harder than base material (3003-H14). It is to say that the solid solution hardening mechanism in the 4043 alloy (Al–Si) is higher than that for the cold worked hardening induced in the 3003-H14 alloy (Al–Mn). It was not observed an evident microhardness decrement in the HAZ for the 3003-H14 alloy, i.e., the mean hardness values close to the WM material remained similar than that of the base material (35 HV0.1). This means that, even when the temperature close to the WM is higher than the annealing temperature for the 3003-H14 alloy (415 °C [20]), the thermal energy was not enough to reach the driving force energy to promote microstructural changes.

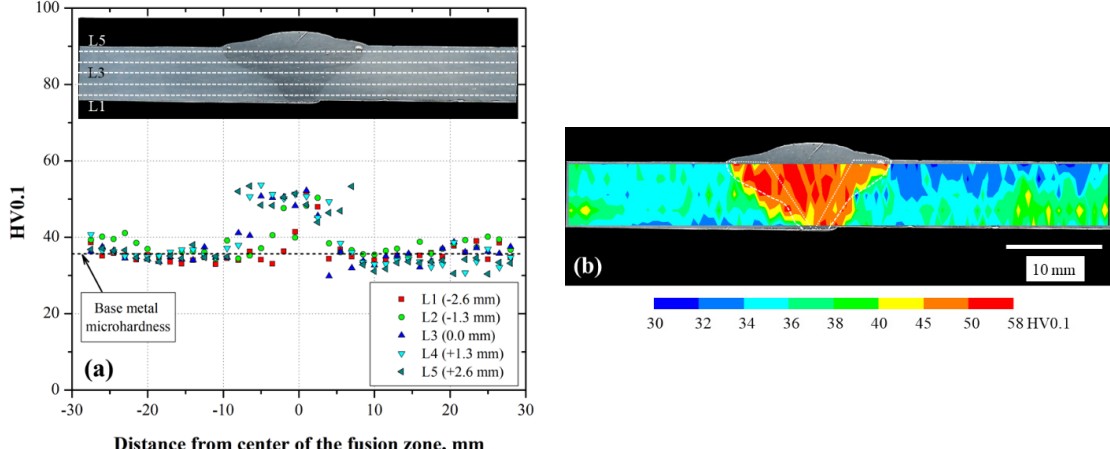

**Figure 8.** Microhardness evolution of the 3003-H14 welded plates: (**a**) microhardness profile and (**b**) map representation.

### 3.2. Finite Element Model

Figure 9 shows the contours of the heat flux distribution in the finite element model for a loading step of 2.75 s. From this figure, it is possible to observe an elliptical heat flux distribution on the surface of the material (weld bead and 3003-H14 aluminum alloy). In addition, it was possible to identify that the heat flux tends to decrease as a function of thickness to reach a magnitude of approximately $10 \times 10^6$ W m$^{-2}$.

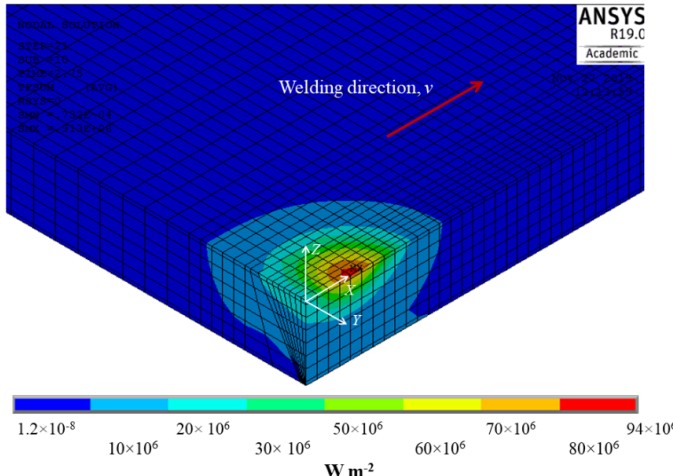

**Figure 9.** Contours of the heat flux distribution according to the double ellipse model (Figure 5). Units are in W m$^{-2}$.

The temperature contours along the welded plates obtained from the application of the heat flux distribution (Figure 9) are shown in Figure 10. It is observed that the maximum temperature in the center of the weld bead is around 2067 °C, at the end of the welding process.

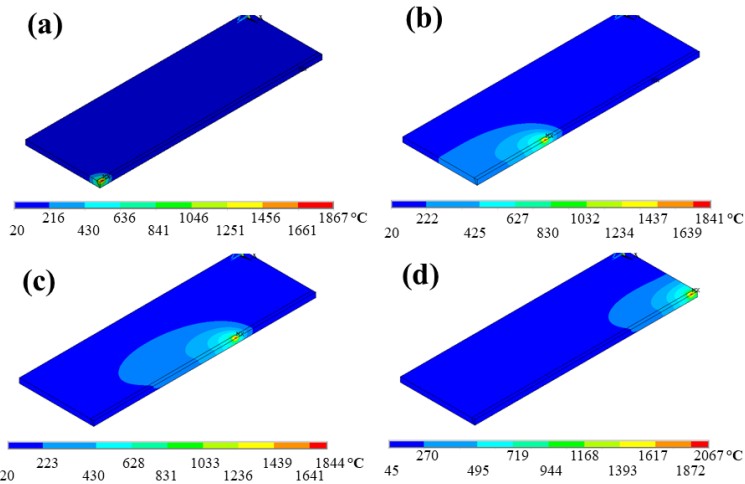

**Figure 10.** Finite element temperature distribution generated by the heat welding source, (**a**) 2.75 s, (**b**) 25.0 s, (**c**) 50.0 s, and (**d**) 75.0 s.

As it is possible to observe in Figure 10c (50 s from the welding starting), the isotherms produced by the heat moving source along the welding process exhibits a well defined elliptical shape. It is not possible to identify a difference in terms of length and width for the isotherms determined at 25 s (100 mm in length) and 50 s (200 mm in length). It is to say that transient heating period seems to be

reached at 25 s. An approximation to the transient time period can be obtained by the application of the dimensionless radius ($\sigma_5$) and time ($\tau$) parameters as suggested by [5]:

$$\sigma_5 = \frac{vr}{2\alpha} \tag{9}$$

$$\tau = \frac{v^2 t}{2\alpha} \tag{10}$$

where $r$ is the distance from the heat source to the point of observation.

By taking $v$ = 4 mm s$^{-1}$, $r$ = 6.5 mm, and $\alpha$ = 60 mm s$^{-2}$, the value for $\sigma_5$ is around 0.22. Thus, according to Figure 11 it is possible to determine that the pseudo-steady state condition is reached when the $\tau$ is approximately 4. This allows to determine a transient heating period of about 30 s, which corresponds to a distance of approximately 120 mm. This approximation was considered to obtain the contours for the fusion (655 °C) and solidification (643 °C) temperatures, as well as the contours of 600 °C, 500 °C, and 400 °C temperatures (Figure 12).

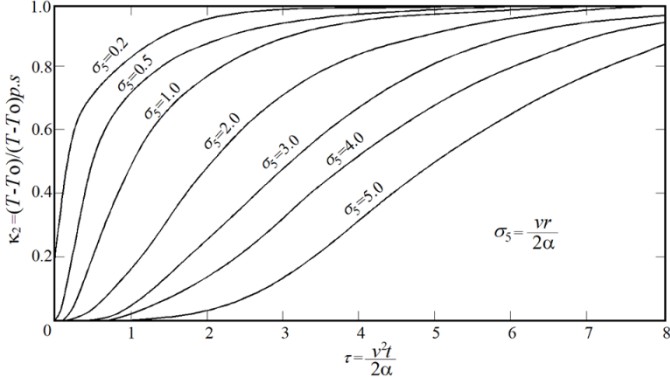

**Figure 11.** Ratio between real and pseudo-steady state temperature in thin plate welding for different combinations of $\sigma_5$ and $\tau$ [5].

The width of the fusion boundary determined by finite element method (Figure 12c) was around 8.3 mm. This value is almost the same as the one determined experimentally by means of the welding profile (8.22 mm). This comparison allows to assume that the used model could determine the weld thermal phenomenon. To observe this assumption, the experimental weld thermal cycles (Figure 13a) were compared with those obtained by finite element method (Figure 13b–d). As it could be observed in Figure 13b–d, the heating and cooling rates as well as the peak temperature are correctly approximated. For instance, a comparison in terms of peak temperature between experimental and finite element results for the $T_1$ and $T_2$ thermocouples revealed an approximation of 2.0%, whereas in the case of the $T_3$ thermocouple is roughly 9.0%. The peak overestimated temperature ($\Delta T$ = 36 °C) for the finite element model at the $T_3$ position of thermocouple, could be attributed to the lack of convective effect by the welding gas, which was only considered for the elements close to the WM.

Once the thermal finite element model was validated on the experimental results, the temperature distribution generated by the heat moving source was used to determine the residual stress field for the welded plates with and without restriction.

To analyze the residual stresses, longitudinal ($\sigma_x$) and transverse ($\sigma_y$) stress distributions were determined. The longitudinal direction (X-direction) is corresponding with the heat moving source, whereas the transverse direction (Y-direction) is perpendicular to the weld bead. Stress contours in longitudinal and transverse direction for the welds without restriction (free displacement) and restricted are shown in Figures 14 and 15.

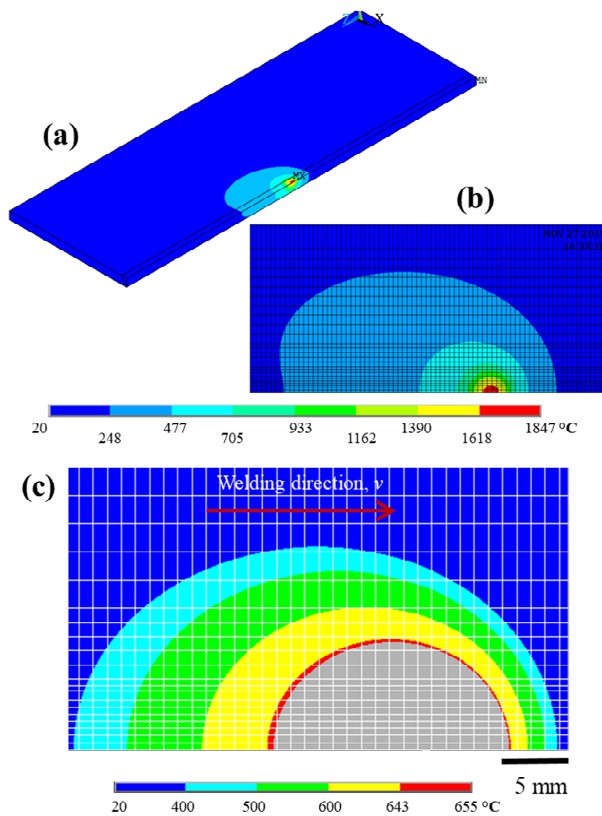

**Figure 12.** Temperature contours in the pseudo-steady state condition during the 3003-H14 welding: (**a**) general view of the isotherms (half model), (**b**) detailed view of the isotherms showed in (**a**), and (**c**) contours for the fusion–solidification temperatures as well as the contours of 600 °C, 500 °C, and 400 °C temperatures.

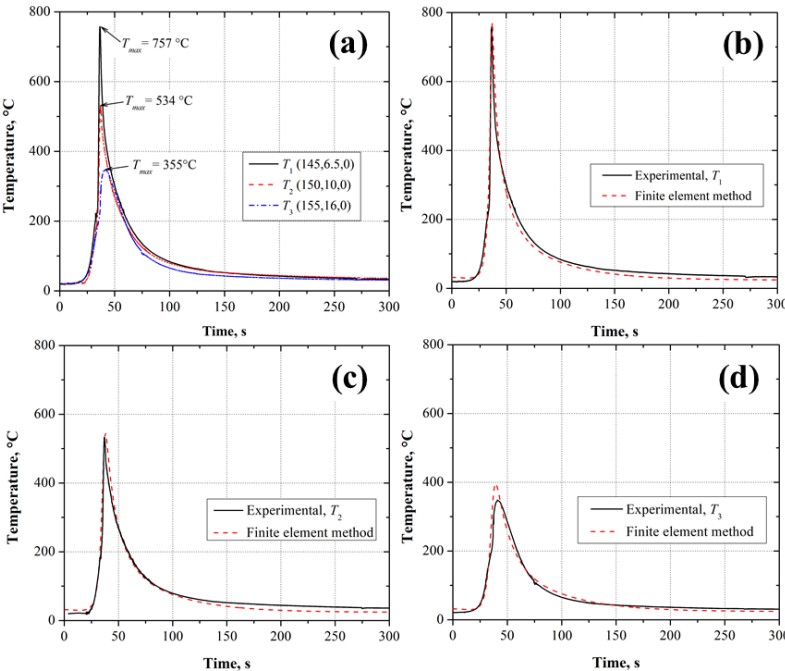

**Figure 13.** (**a**) Experimental weld thermal cycles according to the position of the thermocouples $T_1$, $T_2$, and $T_3$ shown in Figure 1, (**b**–**d**) are the comparison between experimental and finite element results.

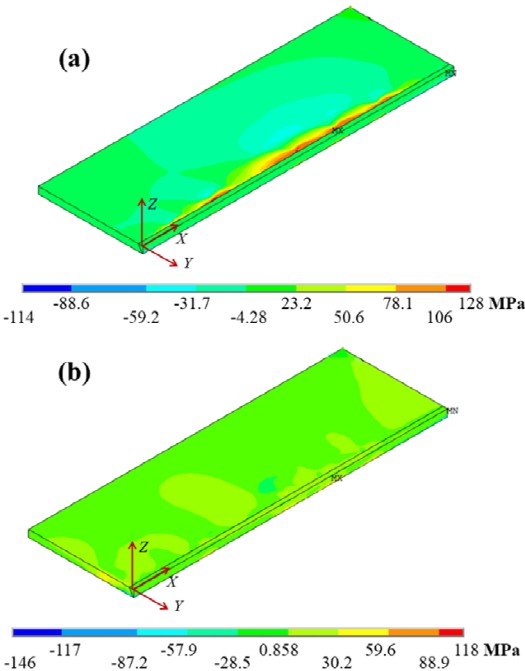

**Figure 14.** Residual stresses generated on the welded plates without restriction: (**a**) X-direction and (**b**) Y-direction.

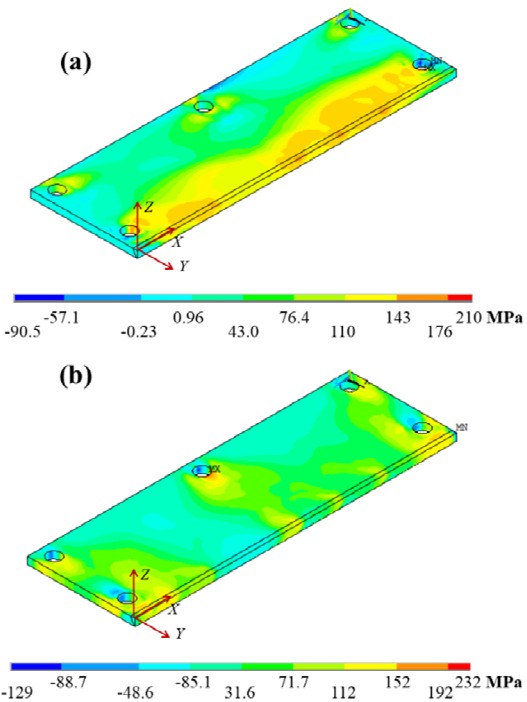

**Figure 15.** Residual stresses generated on the welded plates restricted by bolts: (**a**) X-direction and (**b**) Y-direction.

Tensile and compressive residual stresses were generated in both clamping conditions. The compression (−114 MPa) and tension (128 MPa) residual stresses for the plates without restriction tends to self-balance due to the free degree of freedom. In contrast, the restriction of the plates during welding increases the tensile stresses and tends to decrease the compressive stresses. This fact is produced by the limited ability of the welded plates to the free displacement, i.e., less distortion.

Also, it is possible to observe that residual stresses tend to concentrate close to the fixed zones. To observe the contribution of the axial and shear stresses generated by the thermo-mechanical welding process, von Misses stresses were plotted (Figure 16).

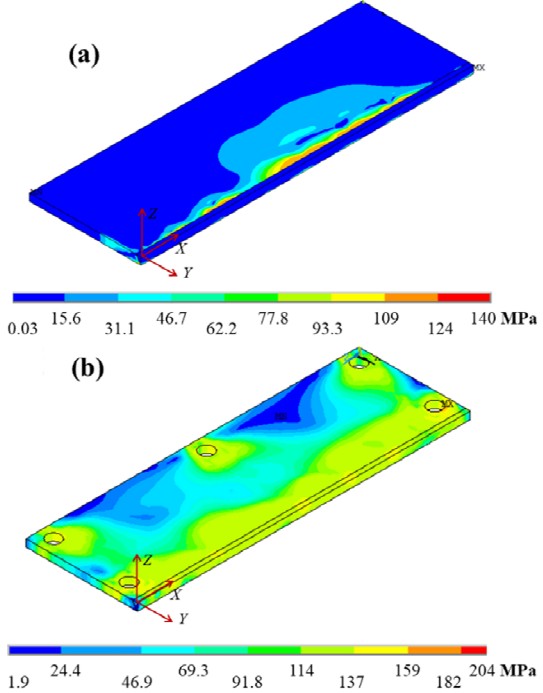

**Figure 16.** Von Mises stress distribution: (**a**) welded plates without restriction and (**b**) welded plates restricted by bolts.

From Figure 16, it is possible to observe that residual stresses concentrate in the surface at the middle of the welded plates without restriction, close to the weld bead. On the other hand, the von Misses stress for the restricted plates increased approximately 46% with respect to the plates without restriction. The residual stresses, in pseudo-steady state condition at the positions shown in Figure 2, were measured by the hole drilling method (Table 4). These values were compared with the ones obtained by the thermo-mechanical model.

**Table 4.** Experimental and finite element results for the longitudinal and transverse residual stresses.

| | Hole Drilling Measurements (MPa) | | | |
|---|---|---|---|---|
| **Strain Rosette** | **Free Condition** | | **Restricted Condition** | |
| | $\sigma_x$ (longitudinal) | $\sigma_y$ (transverse) | $\sigma_x$ (longitudinal) | $\sigma_y$ (transverse) |
| $S_1$ | 76.35 | −17.72 | 112.93 | 133.19 |
| $S_2$ | −5.19 | 20.87 | 42.00 | 88.00 |
| $S_3$ | 25.87 | 19.64 | −0.21 | 121.00 |
| | **Finite Element Results** | | | |
| **Position** | **Free Condition** | | **Restricted Condition** | |
| | $\sigma_x$ (longitudinal) | $\sigma_y$ (transverse) | $\sigma_x$ (longitudinal) | $\sigma_y$ (transverse) |
| $S_1$ | 73.00 | −10.56 | 112.00 | 103.00 |
| $S_2$ | −4.73 | 21.10 | 42.00 | 96.00 |
| $S_3$ | 25.88 | 9.79 | −0.19 | 145.00 |

In general, the finite element (FE) results provide a good estimation of the residual stresses in the welded plates (Table 4). The best estimations are provided for the longitudinal residual stresses with a quite low root mean square error (RMSE) of 1.95 and 0.93 MPa for the unrestricted and restricted clamping conditions. In the case of the transverse residual stresses, the RMSE was 7.03 and 22.74 MPa for the unrestricted and restricted conditions, respectively. The RMSE indicated that the transverse residual stresses were more cumbersome to capture by the proposed FE model. For instance, in the case of the $S_3$ strain rosette position, where the differences between experimental and numerical results can be attributed to a large presence of plastic strain, it could be not correctly captured by the FE model since the fastener bolt deformation was not included in the FE model. It is also relevant to mention that hole drilling is an invasive method, which determines the mean residual stresses by measuring the strain relaxation of a strain rosette placed on the surface of the material, i.e., the strain relaxation measurements are taken as a function of drill depth. This experimental method was used because it is a normalized method, which is available in our laboratory. However, in the future we are considering to determine an approximation of the residual stresses on the surface of the welded plates by X-ray diffraction.

Despite the difference between the experimental and FE results, it was determined that the thermo-mechanical finite element model proposed have an average underestimation of 0.16 with respect to the experimental measured residual stresses after the application of a heat moving source.

Thus, taking into account the FE model, the longitudinal and transverse residual stresses distribution along the surface of the welded plates for three different positions were determined (Figures 17 and 18). In this case, it was only considered the fusion–solidification process, as well as the thermal gradient to analyze the residual stresses. This consideration was taken since the 3003-H14 alloy does not have microstructural transformations in the solid state and taking into account that microhardness profile of the welded joint does not have an important difference in the HAZ.

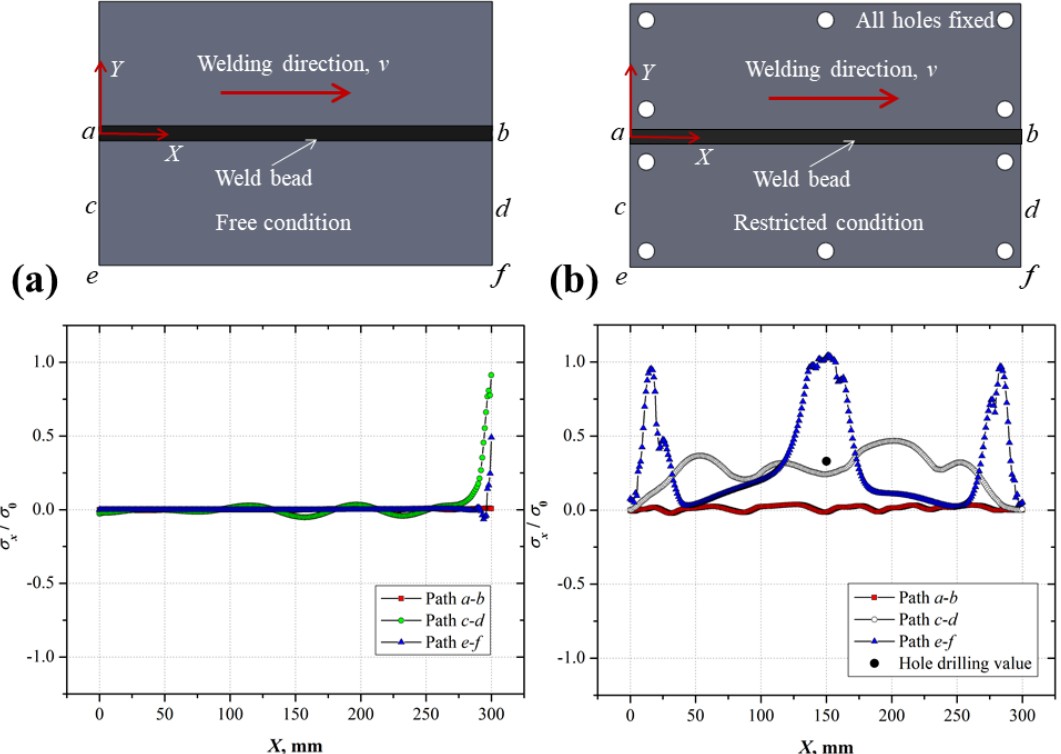

**Figure 17.** Normalized longitudinal residual stresses ($\sigma_x$) as a function of yield stress ($\sigma_0$) of the 3003-H14 aluminum alloy: (**a**) free welding condition and (**b**) restricted condition.

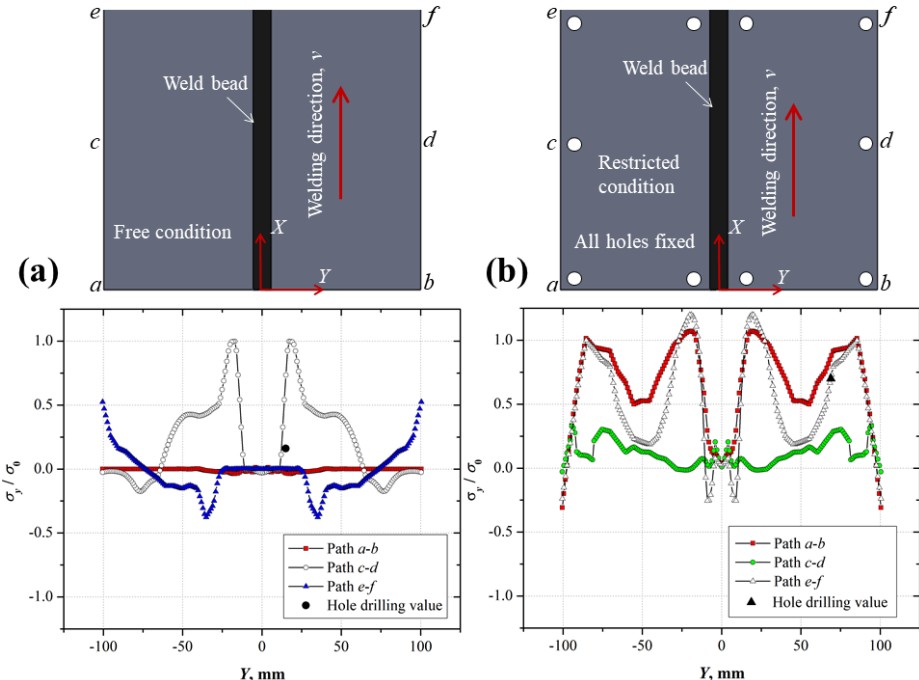

**Figure 18.** Normalized transverse residual stresses ($\sigma_y$) as a function of yield stress ($\sigma_0$) of the 3003-H14 aluminum alloy: (**a**) free welding condition and (**b**) restricted condition.

Figure 17a shows that longitudinal residual stresses for the plates without restriction tends to be zero at the weld bead location (path *a–b* in Figure 17a). This can be associated with a lack of restriction at the center of the weld bead, which is free to expand and contract. However, as a result of the longitudinal contraction-extension (bending) of the welded plate during the solidification, the residual stresses reached a magnitude close to the yield stress of the 3003-H14 alloy along path c–d. Then they decreased at the edge of the plates reaching a stress value close to 42% (path *e–f* in Figure 17a) of the yield stress. A similar behavior (the residual stresses tends to be zero) was observed for the longitudinal residual stresses along the weld bead of the restricted plates (path *a–b* in Figure 17b). However, due to the restricted condition during and after the welding process an increment of the residual stresses were observed along the paths *c–d* and *e–f* (Figure 17b). From the path *c–d*, which is approximately at the middle of a welded plate along the Y-direction, it is possible to observe that the stress distribution has a wavy shape reaching a maximum normalized value of about 0.5 (longitudinal residual stress $\sigma_x$ versus yield stress $\sigma_0$ of the 3003-H14 alloy) at a distance of 200 mm from the starting of the heat moving source. Regarding the path *e–f*, due to the high restriction set by the fastened bolts close to the edge of the welded plates, the residual stresses increased considerably (stress concentration effect). The $\sigma_x/\sigma_0$ values tend to be similar than yield stress of the 3003-H14 alloy.

The normalized transverse residual stresses ($\sigma_y$) as a function of yield stress ($\sigma_0$) for the 3003-H14 aluminum alloy are shown in Figure 18.

With respect to the free clamping condition (Figure 18a), it is possible to observe that heat moving source produced different residual stress distributions (paths *a–b*, *c–d*, and *e–f*). This phenomenon is produced by the localized thermal strain induced by the welding process, i.e., a very high thermal gradient produced by the fusion–solidification process as well as the heat transient conduction phenomenon towards the right and left zones of the weld bead. In addition, the welded bead produced by the heat moving source modified the clamping condition in the welded plate. From a free condition at the starting of the welding process (path *a–b*) to a partial restricted condition towards the end of the welding process (path *e–f*). This explains why the transverse residual stresses were almost zero for the path *a–b*, because the welded plate was completely free to expand and contract during the fusion-solidification process. As possible to observe in Figure 18a, the solidified region presented

residual stresses close to zero, which was like the case of the longitudinal residual stresses (Figure 17a). In this context, it is possible to say that the partial restriction set by the weld bead induced tensile residual stresses close to the WM for path *c–d*, up to a width of about 60 mm. Later, the residual stress distribution went down to negative values to finally increased gradually to reach values close to zero. In the case of the end of the welding process (path *e–f*), the increment in the accumulative heat and temperature induced to the welded plate seems to be responsible for the wider region of residual stresses close to a zero value. Also, a compressive residual stresses zone was produced, but with lower intensity with respect to the tensile residual stresses presented along path *c–d*. Finally, due to the pseudo-state thermal conduction as well as the partial restriction set by the weld bead, tensile residual stresses emerged towards the edge of the welded plate (path *e–f*).

For the case of the welded plate with the restricted condition (Figure 18b), different residual stress distributions were also produced by the heat moving source (paths *a–b*, *c–d*, and *e–f*). As well, the solidified region presented residual stresses close to a zero value, but over a quite small region with respect to the free condition. However, zero residual stresses results in the WM of the welded plate with the different clamping conditions should be considered with caution, because the FE model did not include a material model for the liquid–solid transformation of the filler material, and it is likely that a low value of residual stresses will be presented in the WM. Overall, the transverse residual stresses produced by the restricted condition presented the largest values with respect to the free condition (Figure 18a). Paths *a–b* and *e–f* in the transverse direction exhibited the most critical residual stresses distribution in the welded component. The critical position was at the interface between WM and BM regions, where the residual stress value was larger than the yield stress of the 3003-H14 aluminum alloy. This critically conditions were produced by the full restriction set by the fastener bolts, which prevented the free expansion and contraction movement of the welded plate.

## 4. Conclusions

The thermo-mechanical analysis conducted by the three-dimensional finite element model was able to capture the welding thermal history experienced by the 3003-H14 aluminum plates under two thermal expansion conditions (restricted and free). In addition, welding residual stresses determined by the finite element model exhibited a good approximation with respect to experimental measurements.

Heating and cooling rates as well as peak temperatures and the pseudo-steady state temperature contours of the heat moving source obtained numerically were in good agreement with the experimental results. A temperature difference of just 2.0% and 9.0% was observed between the finite element and experimental results.

From the finite element model, it was observed that regardless of the restriction condition, the longitudinal residual stresses along the weld bead tend to be zero. For the restricted condition it was observed that longitudinal and transverse normal residual stresses tend to be in tension, and they changed as a function of length and width of the welded plates. The most critical residual stresses were present in the transverse welding direction for both thermal expansion conditions.

**Author Contributions:** Conceptualization, R.R.A. and D.J.; methodology, M.H., R.R.A.; Software, M.H.; validation, M.H., R.R.A., C.G. and D.J.; writing-review and editing, M.H., R.R.A., C.G. and D.J.; supervision, R.R.A. and D.J.; funding acquisition, R.R.A. and D.J. All authors have read and agreed to the published version of the manuscript.

**Funding:** This research was funded by CONACyT-México (project A1-S-27474) and SIP-IPN.

**Acknowledgments:** The authors thank to CONACyT-México and SIP-IPN.

**Conflicts of Interest:** The authors declare no conflict of interest.

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
