# Peer review of "The Thermomechanical Finite Element Analysis of 3003-H14 Plates Joined by the GMAW Process"

_metals, doi:10.3390/met10060708_

Round 1

Reviewer 1 Report

The submission is certainly interesting and worth publishing, however some points have to be considered before:

  1. The reader would like to see a close-up of the strain rosette.
  2. The values given for a_f and a_r are in contradiction to Fig.4.
  3. In line 178 it is referred to Fig. 8; it should be Fig. 7.
  4. The typesetting of the symbols is different in the list and the equations.
  5. The setting of the references is inconsistent.
  6. There are quite a few English mistakes and some typos.

Author Response

Point 1: The reader would like to see a close-up of the strain rosette.

Response 1: Figure 2 has been modified to include a close-up of the strain rosette used to measure residual stresses close to the weld metal.

Point 2: The values given for a_f and a_r are in contradiction to Fig.4.

Response 2: The correct values according to Figure 4 and experimental measurements for af and ar were changed. Thank you for your observation.

Point 3: In line 178 it is referred to Fig. 8; it should be Fig. 7.

Response 3: This mistake was emended. Please verify this correction on the manuscript highlighted in yellow.

Point 4: The typesetting of the symbols is different in the list and the equations.

Response 4: List of symbols has been eliminated according to the suggestion of Reviewer 2. Also, the typesetting of the symbols has been verified to be in accordance with equations.

Point 5: The setting of the references is inconsistent.

Response 5: References were verified and written according to the format asked by the journal.

Point 6: There are quite a few English mistakes and some typos.

Response 6: The English has been carefully verified according to our knowledge. Some typos were emended.

Reviewer 2 Report

The authors have proposed an interesting paper but prior to publication, the following points must be addressed:

  1. Line  - I suggest deleting the list of symbols and explaining where they will be used for the first time.
  2. Line 94 - justify the choice of filler material in terms of the weldability of the base material.
  3. Line 102 - gas flow is given in l/min
  4. Line 105 - the thermal efficiency coefficient is according to EN 1011-2 for the GMAW (131/135) welding method k = 0.8
  5. Line 116 - The NI 9213 module has a maximum sampling frequency equal 75Hz. The value of 200Hz is given in the article. Is this correct?
  6. Line 125 - unit is missing
  7. Line 108 - The text describes the hardness HV0.1, while in the graph in fig. 8 is HV0,2 on the y-axis. What is correct?
  8. Vickers hardness measurement is non-standard for Al-alloy welds. Why measure hardness in multiple lines? You did not use this information for modeling.
  9. Line 147 - FZ (Fusion zone) is incorrect designation. Correct is WM (Weld metal)
  10. Line 335 - The table number should be in Latin, not Roman numerals.

Further notes on the article:

  1. The article lacks data on the change in mechanical characteristics of the base and additive material for temperatures up to the melting point range.
  2. The course of residual stresses (Figs. 17 and 18) is for what depth in the materials? Is it a surface? It is not clear.
  3. The drilling depth is not described in the experimental measurement of residual stresses.

Author Response

Point 1: Line  - I suggest deleting the list of symbols and explaining where they will be used for the first time.

Response 1: The list of symbols has been eliminated. An explanation of each symbol is included where they appear by first time.

Point 2: Line 94 - justify the choice of filler material in terms of the weldability of the base material.

Response 2: The 3003 aluminum alloy does not have problems in terms of weldability. In fact, this aspect has been verified by means of microhardness profiles measurements. A commercial filler material such as ER4043 was selected by considering the guide to the choice of filler metal for general purpose welding (Table 2 in AWS A5.10).

Point 3: Line 102 - gas flow is given in l/min

Response 3: The units have changed according to the recommendation.

Point 4: Line 105 - the thermal efficiency coefficient is according to EN 1011-2 for the GMAW (131/135) welding method k = 0.8

Response 4: Different values have been reported for the thermal efficiency. For instance J. N. DuPont and A. R. Marder [1] report that thermal efficiency is a function of the welding current. They reported values between 0.75 to 0.9 for a GMAW process. In this context, in 2009 we conducted a study and it was found that thermal efficiency is about 0.9 for a 6061-T6 aluminum alloy [2]. We have been used this value previously with an acceptable accuracy [3, 4].

  1. Thermal efficiency of arc welding processes. J. N. DuPont and A. R. Marder. Supplement of the American Welding Journal, Vol. 11, pp. 406-416, 1995.
  2. Thermal efficiency in welding of AA6061-T6 alloy by modified indirect electric arc and digitalization of current signals. R. R. Ambriz, G. Barrera, R. García, V. H. López. Welding International, Vol. 25, No. 2, pp. 86-93, 2011. ISSN 0950-7116 (print), ISSN 1754-2138 (online). https://doi.org/10.1080/09507116.2010.527052
  3. Heat distribution in welds of a 6061-T6 aluminum alloy obtained by modified indirect electric arc. C. M. Gómora, R. R. Ambriz, F. F. Curiel, D. Jaramillo. Journal of Materials Processing Technology, Vol. 243, No. 5, pp. 433-441, 2017. ISSN: 0924-0136. https://doi.org/10.1016/j.jmatprotec.2017.01.003
  4. Heat sink effect of 6061 aluminum alloy welds with different partial ageing conditions. C. M. Gomora, R. R. Ambriz, J. Zuno-Silva, D. Jaramillo. Submitted to Journal of Materials Engineering and Performance.

Point 5: Line 116 - The NI 9213 module has a maximum sampling frequency equal 75Hz. The value of 200Hz is given in the article. Is this correct?

Response 5: There was a mistake, the correct frequency at which the temperature measurements were taken was 75 Hz. This aspect has been amended.  

Point 6: Line 125 - unit is missing

Response 6: Are you referring to Figure 2a? A text in the Figure caption was included to specify that units in Figure 2 are in mm. 

Point 7: Line 108 - The text describes the hardness HV0.1, while in the graph in fig. 8 is HV0,2 on the y-axis. What is correct?

Response 7: Thank you for this observation. The load used for microhardness measurements was 0.1 kg. Figure 8 has been replaced.

Point 8: Vickers hardness measurement is non-standard for Al-alloy welds. Why measure hardness in multiple lines? You did not use this information for modeling.

Response 8: In the literature, there are many articles, which evaluate the Vickers microhardness in welded joints for different materials including aluminium alloys. Also, these measurements have used to plot hardness profiles (multiple lines) and mapping representation. The microhardness information in this case was not used in the finite element model. It was used to evaluate the heat effect produced by the heat moving source.

Point 9: Line 147 - FZ (Fusion zone) is incorrect designation. Correct is WM (Weld metal)

Response 9: The phrase fusion zone was changed by weld metal.

Point 10: Line 335 - The table number should be in Latin, not Roman numerals.

Response 10: Thank you for your observation, the table number has been changed.

Further notes on the article:

Point 1: The article lacks data on the change in mechanical characteristics of the base and additive material for temperatures up to the melting point range.

Response 1: Sorry, we do not have this information to include in the article.

Point 2: The course of residual stresses (Figs. 17 and 18) is for what depth in the materials? Is it a surface? It is not clear.

Response 2: Figure 17 and 18 report the residual stress distribution on the surface of the welded plates. Line 354 has been changed to clarify this.

Point 3: The drilling depth is not described in the experimental measurement of residual stresses.

Response 3: Lines 134 and 135 described the drilling depth.

Round 2

Reviewer 2 Report

Please make small corrections:

1. The hardness mark is correctly HV 0.1. Subscript is not used. Change in the text and in Figure 8.

2. Figure 8. It would be advisable to change the type of graph to a hardness map. See this link for an example:

https://www.researchgate.net/figure/Hardness-maps-from-a-FSW-and-b-SSFSW-cross-sections-measured-after-six-months_fig6_310319715

The graph in this form has no informative value.

3. Your citation from previous review: "The microhardness information in this case was not used in the finite element model. It was used to evaluate the heat effect produced by the heat moving source."

My answer: Alloy 3003 is not a hardenable aluminum alloy. Therefore, the heat load will not have a significant effect on the change in hardness. Your results confirm this.

4. My note from a previous review "Point 6: Line 125 - unit is missing"

My answer:Sorry, I meant line 128, the correct unit was missing, you have already added it. The description of Figure 2 was good.

Author Response

Point 1: The hardness mark is correctly HV 0.1. Subscript is not used. Change in the text and in Figure 8.

Response 1: Thank you for your observation. The subscript was eliminated in Figure 8 as well as in the text.

Point 2: Figure 8. It would be advisable to change the type of graph to a hardness map. See this link for an example:

https://www.researchgate.net/figure/Hardness-maps-from-a-FSW-and-b-SSFSW-cross-sections-measured-after-six-months_fig6_310319715

The graph in this form has no informative value.

Response 2: A map representation has been included in Figure 8.

Point 3: Your citation from previous review: "The microhardness information in this case was not used in the finite element model. It was used to evaluate the heat effect produced by the heat moving source."

My answer: Alloy 3003 is not a hardenable aluminum alloy. Therefore, the heat load will not have a significant effect on the change in hardness. Your results confirm this.

Response 3: Yes as suggested, the results confirm this fact.

Point 4: My note from a previous review "Point 6: Line 125 - unit is missing"

My answer:Sorry, I meant line 128, the correct unit was missing, you have already added it. The description of Figure 2 was good.

Response 4: Yes, it has been corrected.
